# Untargeted Metabolomics Profiling Reveals Beneficial Changes in Milk of Sows Supplemented with Fermented Compound Chinese Medicine Feed Additive

**DOI:** 10.3390/ani12202879

**Published:** 2022-10-21

**Authors:** Wanjie Zou, Linglan Deng, Huadong Wu, Zhiyong Liu, Wei Lu, Yuyong He

**Affiliations:** 1Jiangxi Province Key Laboratory of Animal Nutrition/Engineering Research Center of Feed Development, Jiangxi Agricultural University, Nanchang 330045, China; 2College of Animal Science and Technology, Jiangxi Agricultural University, Nanchang 330045, China; 3Key Laboratory of Pharmacology of Traditional Chinese Medicine in Jiangxi, Jiangxi University of Traditional Chinese Medicine, Nanchang 330004, China

**Keywords:** fermented Chinese medicine feed additive, differential metabolites, milk, sow, untargeted metabolomics approach

## Abstract

**Simple Summary:**

Diarrhea often occurs in suckling piglets and milk secreted by lactating sows with metritis–vaginitis–mastitis is one of the most important contributors. Chinese herbal medicine has antibacterial and anti-inflammatory effects due to its bioactive ingredients, and it is of interest to explore whether maternal feeding with Chinese herbal medicine can increase the level of milk ingredients with anti-infectious and anti-inflammatory properties. The present study found that supplementation of fermented compound Chinese medicine feed additive to sows increased the concentration of functional ingredients, such as quercetin, pinocembrin, chlorogenic acid, methyl succinic acid, L-tryptophan, adenosine, guanine, arteannuin, inosine, guanosine, benzene-1,2,4-triol, hypoxanthine, adenine, ferulic acid, echimidine N-oxide, pogostone, kynurenine and trehalose 6-phosphate in the milk of sows, most of these functional ingredients have anti-infectious, anti-inflammatory, anti-oxidative and immune-enhancing effects. The findings of this study hint that supplementation with fermented compound Chinese medicine feed additive in sows is beneficial for the improvement of milk quality.

**Abstract:**

Different untargeted metabolomics approaches were used to identify the differential metabolites between milk samples collected from two groups. Sows were supplemented with fermented compound Chinese medicine feed additive at levels of 0 g/d/sow (control group, *n* = 10) and 50 g/d/sow (experimental group, *n* = 10), respectively, from d 104 of gestation to d 25 of lactation, samples of colostrum and mature milk were collected. Data indicated that supplementing fermented compound Chinese medicine feed additive to sows significantly increased the concentrations of quercetin, pinocembrin, chlorogenic acid, methyl succinic acid, L-tryptophan, adenosine, guanine, arteannuin, ferulic acid, echimidine N-oxide, pogostone and kynurenine in the colostrum and inosine, guanosine, benzene-1,2,4-triol, hypoxanthine, adenine, trehalose 6-phosphate in mature milk, respectively. Seven pathways (flavone and flavanol biosynthesis, galactose metabolism, phenylpropanoid biosynthesis, stilbenoid and gingerol biosynthesis, flavonoid biosynthesis, ABC transporters and purine metabolism) in colostrum and two pathways (sucrose metabolism and retrograde endocannabinoid signaling) in mature milk were significantly enriched in the experimental group compared to control group, respectively. The supplementation of fermented compound Chinese medicine feed additive to sows increased the level of antibacterial and anti-inflammatory ingredients in milk and the findings of this study hint that supplementation with fermented compound Chinese medicine feed additive in sows is beneficial for the improvement of milk quality.

## 1. Introduction

Sow’s milk is a complex biological fluid and contains macro-chemicals, micro-chemicals and microbes. The composition of sow’s milk often undergoes alterations during the lactation period with the changes in breeds, parities, diets and disease status [1,2,3,4,5,6,7]. It is well known that milk is the main food for piglets during the suckling period, so the quality of sow milk has a vital impact on the survival and growth of suckling piglets [8,9,10]. There are many methods to improve the quality of sow’s milk including the use of different feeding regimens [11] or plant-derived bioactive compounds [12,13,14].

Metritis–vaginitis–mastitis of sows is one of the most prevalent and costly diseases in pig farms, milk secreted by sows with metritis–vaginitis–mastitis often contains harmful microbes and pro-inflammatory mediators, and ingestions of mastitis milk and fecal pathogens from sows are the major contributors to the high mortality of suckling piglets owing to severe diarrhea which is caused by the pathogens, allergic substances and pro-inflammatory mediators in the guts of suckling piglets [3,4,15], this is the reason why sows often have a high number of piglets born alive but a low number of piglets at weaning.

Chinese herbal medicines or extracts are often added to the diet to promote the healthy production of cow’s milk [16,17] and sow’s milk [18] because Chinese herbal medicines and extracts contain bioactive components that possess antibacterial, anti-inflammatory, anti-oxidative and immune-enhancing properties [19,20,21,22]. Some of these bioactive compounds can be transferred directly from Chinese medicine into milk, and some of them can be metabolized to other bioactive compounds by gut microbes and then enter into the milk; providing this functional milk with high levels of Chinese herb medicine-derived bioactive ingredients to suckling piglets is one of the methods to decrease diarrhea and mortality in neonates. In the previous study, we found that fermented compound Chinese medicine feed additive can effectively inhibit the growth of some bacteria associated with metritis–vaginitis–mastitis due to its bioactive metabolites [23]. Chinmedomics strategy is an integration of serum pharmco-chemistry of traditional Chinese medicine and “Omics” technology and is often performed to determine the components of Chinese medicine [24,25]; a metabolomics approach is usually applied to find out the non-Chinese medicine originated compounds [26]. In the present experiment, we applied different untargeted metabolomics approaches to characterize whether maternal feeding with fermented compound Chinese medicine feed additive has an impact on the composition of active ingredients in milk.

## 2. Materials and Methods

### 2.1. Animals and Feeding

This feeding experiment started on 20 August 2021, and twenty pregnant crossbred sows (Landrace × Large White) with similar body conditions and parities were randomly assigned to the control group and experimental group with 10 sows (10 replicates) in each group according to a randomized complete block design and fed the same basal diet (Table 1) with supplementation of fermented compound Chinese medicine feed additive from d 104 of gestation to d 25 of lactation at doses of 0 and 50 g/sow, respectively, once a day in the morning. Prior to feeding, the allowance of morning meal of each sow was divided into two parts, one part was mixed with the fermented compound Chinese medicine feed additive and then offered to the sow, and the other part was followed after the sow consumed the mixture of meal and fermented compound Chinese medicine feed additive. The information on ingredients, preparation and chemical compositions of fermented compound Chinese medicine feed additive have been reported in a published paper [23].

### 2.2. Sample Collection and Preparation

Samples of milk were collected with sterile Eppendorf tubes (Hamburg, Germany), and samples of colostrum and mature milk from each sow were daily collected from d 1 to 5 and d 10 to 20 relative to parturition, respectively, all samples were stored at −20 °C. At the end of sampling, samples of colostrum and mature milk of each sow were mixed, respectively, and the mixture of samples was sub-packed with 5 mL sterile Eppendorf tubes and then stored at −20 °C before analysis.

### 2.3. Analysis of Active Ingredients Using UHPLC-QE-MS Based Untargeted Chinmedomics

Samples of milk were processed for the extraction of Chinese medicine active ingredients according to the standardized protocols (Shanghai Biotree Biotech Co., Ltd., Shanghai, China) before analysis. LC-MS/MS analysis was performed using a 1290 UPLC system (Agilent, Santa Clara, CA, USA) with a Waters UPLC BEH C18 column (1.7 μm, 2.1 × 100 mm). The flow rate was set at 0.4 mL/min and the sample injection volume was set at 5 μL. The mobile phase consisted of 0.1% formic acid in water (A) and 0.1% formic acid in acetonitrile (B). The multi-step linear elution gradient program was as follows: 0–3.5 min, 95–85% A; 3.5–6 min, 85–70% A; 6–6.5 min, 70–70% A; 6.5–12 min, 70 -30% A; 12–12.5 min, 30–30% A; 12.5–18 min, 30–0% A; 18–25 min, 0–0% A; 25–26 min, 0–95% A; 26–30 min, 95–95% A.

A Q Exactive Focus mass spectrometer (Vanquish, Thermo Fisher Scientific, Waltham, MA, USA) coupled with Xcalibur TM 3.0 software (Thermo Fisher, Waltham, MA, USA) was employed to obtain the MS and MS/MS data based on the information-dependent acquisition mode. During each acquisition cycle, the mass range was from 100 to 1500, the top three of every cycle were screened and the corresponding MS/MS data were further acquired. The ESI source was applied to analyze the chemical composition in both positive and negative ion modes with full scan/ddMS2. Sheath gas flow rate: 45 Arb, Aux gas flow rate: 15 Arb, Capillary temperature: 400 °C, Full MS resolution: 70,000, MS/MS resolution: 17,500, Collision energy: 15/30/45 in NCE mode, Spray Voltage: 4.0 kV (positive) or −3.6 kV (negative).

Raw data were processed using the XCMS package, and the qualified data were uploaded to SIMCA-P (Version 16.0.2, Sartorius Stedim Data Analytics AB, Umea, Sweden) for statistical analysis. Significantly altered metabolites were determined by *t*-test and a *p* value < 0.05 was considered statistically significant. Identification of active ingredients was performed by searching the Biotree databases and web databases (METLIN, HMDB, PubChem, and ChemSpider).

### 2.4. Identification of Differential Metabolites Using UHPLC-QE-MS Based Conventional Untargeted Metabolomics

Samples of milk were processed according to the standardized protocols before UHPLC-QE-MS-based conventional untargeted metabolomic analysis. LC-MS/MS analyses were performed using a 3000 UHPLC system (Vanquish, Thermo Fisher Scientific, Waltham, MA, USA) with a UPLC HSS T3 column (1.8 μm, 2.1 mm × 100 mm) coupled to a Q Exactive HFX mass spectrometer (Orbitrap MS, Thermo, Waltham, MA, USA). The mobile phase consisted of 5 mmol/L ammonium acetate and 5 mmol/L acetic acid in water (A) and acetonitrile (B). The auto-sampler temperature was 4 °C, and the injection volume was 2 μL. The QE HFX mass spectrometer was used for its ability to acquire MS/MS spectra on information-dependent acquisition mode in the control of the acquisition software (Xcalibur TM 3.0, Thermo, Waltham, MA, USA). In this mode, the acquisition software continuously evaluates the full scan MS spectrum. The ESI source conditions were set as follows: sheath gas flow rate as 30 Arb, Aux gas flow rate as 10 Arb, capillary temperature 350 °C, full MS resolution as 60,000, MS/MS resolution as 7500, collision energy as 10/30/60 in NCE mode, spray Voltage as 4.0 kV (positive) or −3.8 kV (negative), respectively.

Raw data were converted to the mzXML format using ProteoWizard and processed with an in-house program, which was developed using R and based on XCMS for peak detection, extraction, alignment and integration. Significantly altered metabolites were determined by *t*-test and a *p* value < 0.05 was considered statistically significant. An in-house MS2 database (Biotree database) was applied in metabolite annotation and the cutoff for annotation was set at 0.3.

## 3. Results

### 3.1. Differential Metabolites in Colostrum between Experimental and Control Groups Based on Chinmedomics

Twenty-six metabolites under the negative ion model and twenty-two metabolites under the positive ion model were identified in colostrum between experimental and control groups including eight flavonoids, five phenols, eight alkaloids, two phenylpropanoids, eight terpenoids, two fatty acyls, two fatty acids, one organoheterocyclic compound, two organic acids and derivatives, one organic oxygen compound, six amino acid derivatives, one carbohydrate and derivative, one aliphatic and one organooxygen compound (Table 2). The concentrations of 48 differential metabolites in the colostrum of the experimental group were numerically or significantly higher than that in the colostrum of the control group, and the colostrum of the experimental group had significantly higher concentrations of quercetin (*p* < 0.05), pinocembrin (*p* < 0.05), chlorogenic acid (*p* < 0.01), methyl succinic acid (*p* < 0.01), L-tryptophan (*p* < 0.01), adenosine (*p* < 0.05), guanine (*p* < 0.05) and arteannuin (*p* < 0.05) compared to the colostrum of the control group, respectively.

### 3.2. Differential Metabolites in Mature Milk between Experimental and Control Groups Based on Chinmedomics

Table 3 showed that a total of 32 metabolites were screened under positive and negative ion modes, respectively, in mature milk between the experimental and control groups including four phenols, seven phenylpropanoids, one xanthone, two sesquiterpenoids, three amino acid derivatives, four alkaloids, three flavonoids, one chalcone, four terpenoids, one organooxygen compound, one fatty acid and one carboxylic acid and derivative. The mature milk of the experimental group had numerically or significantly higher concentrations of 32 metabolites compared to the mature milk of the control group, the concentrations of ferulic acid (*p* < 0.05), echimidine N-oxide (*p* < 0.05), pogostone (*p* < 0.05) and kynurenine (*p* < 0.05) in the mature milk of the experimental group were significantly higher than that of the control group, respectively.

### 3.3. Differential Metabolites between Colostrum and Mature Milk of Experimental Group Based on Chinmedomics

A total of 23 metabolites were identified under positive and negative ion modes between the colostrum and mature milk in the experimental group including one flavonoid, one phenol, eight alkaloids, seven terpenoids, two phenylpropanoids, one coumarin and derivative, one phospholipid, one organoheterocyclic compound and one fatty acid (Table 4). Colostrum had significantly higher concentrations of bergenin (*p* < 0.05), 3-furfuryl 2-pyrrolecarboxylate (*p* < 0.05), guanosine (*p* < 0.01), guanine (*p* < 0.05), palmatine (*p* < 0.01), celastrol (*p* < 0.05), lindenenol (*p* < 0.05), artemisinin (*p* < 0.05), curcumenol (*p* < 0.05) and aucubin (*p* < 0.05) than mature milk in the experimental group, respectively. 

### 3.4. Differential Metabolites between Colostrum and Mature Milk of Control Group Based on Chinmedomics

A total of 22 metabolites were found under positive and negative ion modes between colostrum and mature milk in the control group including one organic acid and derivative, one phenol, eight alkaloids, six terpenoids, two phenylpropanoids, one coumarin and derivative, one organooxygen compound, one benzene and substituted derivative and one amino acid derivative (Table 5). Colostrum had significantly higher concentrations of threonic acid (*p* < 0.05), bergenin (*p* < 0.01), boldine (*p* < 0.05), 3-Furfuryl 2-pyrrolecarboxylate (*p* < 0.05), aucubin (*p* < 0.05), celastrol (*p* < 0.01) and eudesmin (*p* < 0.01) than mature milk in the control group, respectively. 

### 3.5. Differential Metabolites in Colostrum between Experimental and Control Groups Based on Conventional Untargeted Metabolomics

A total of seventy-six differential metabolites have been identified in the colostrum between experimental and control groups including 24 organic acids and derivatives, 8 organoheterocyclic compounds, 1 organooxygen compound, 4 organic oxygen compounds, 1 organic nitrogen compound, 11 nucleosides, nucleotides, and analogs, 21 lipids and lipid-like molecules and 6 benzenoids (Table 6). The colostrum of sows in the experimental group had significantly higher levels of inosine, guanosine, benzene-1,2,4-triol, hypoxanthine and adenine than the colostrum of sows in the control group (*p* < 0.05), respectively. Other metabolites in the colostrum of the experimental group also had numerically higher concentrations than that in the colostrum of the control group (*p* > 0.05), respectively.

### 3.6. Differential Metabolites in Mature Milk between Experimental and Control Groups Based on Conventional Untargeted Metabolomics

Table 7 indicated that 13 differential metabolites under the negative ion model and 10 differential metabolites under the positive ion model had been screened in mature milk between the experimental group and control group including seven organic acids and derivatives, two organoheterocyclic compounds, three organic oxygen compounds, one organic nitrogen compound, three nucleosides, nucleotides and analogs, five lipids and lipid-like molecules and two benzenoids. The concentration of trehalose 6-phosphate in the mature milk of the experimental group was significantly higher than that in the mature milk of the control group (*p* < 0.05), and other metabolites in the mature milk of the experimental group had numerically higher levels than those in the mature milk of the control group (*p* > 0.05), respectively.

### 3.7. Differential Metabolites between Colostrum and Mature Milk of Experimental Group Based on Conventional Untargeted Metabolomics

Nineteen differential metabolites under the negative ion model and thirty-two differential metabolites under the positive ion model had been found in the experimental group between colostrum and mature milk including 11 organic acids and derivatives, 5 organic oxygen compounds, 6 organoheterocyclic compounds, 1 organohalogen compound, 18 lipids and lipid-like molecules, 9 nucleosides, nucleotides and analogs and 1 benzenoid (Table 8). Thirty-five differential metabolites had significantly higher concentrations in colostrum than in mature milk and the other sixteen metabolites in colostrum also had numerically higher levels than in mature milk. 

### 3.8. Differential Metabolites between Colostrum and Mature Milk of Control Group Based on Conventional Untargeted Metabolomics

Thirty-seven differential metabolites had been screened between colostrum and mature milk in the control group including 5 organic acids and derivatives, 12 organic oxygen compounds, 4 organoheterocyclic compounds, 1 organohalogen compound, 8 nucleosides, nucleotides and analogs, and 7 lipids and lipid-like molecules (Table 9). Twenty-seven metabolites in colostrum had significantly higher concentrations than that in mature milk and ten metabolites had numerically higher levels in colostrum than in mature milk. 

### 3.9. Metabolic Pathways Enrichment

In order to identify the changes in the metabolic pathway reflected by differential metabolites, differential abundance analysis of KEGG metabolic pathways for differential metabolites screened by chinmedomics was conducted and results are shown in Figure 1. There were significant differences in the differential abundances of five metabolic pathways of differential metabolites in the colostrum between the experimental group and control group (Figure 1A), differential metabolites enriched in the pathways of flavone and flavanol biosynthesis, galactose metabolism, phenylpropanoid biosynthesis, stilbenoid and gingerol biosynthesis, flavonoid biosynthesis were upregulated in the colostrum of sows supplemented with fermented compound Chinese medicine feed additive. Two pathways of differential metabolites in mature milk had significant differences in differential abundance (Figure 1B), and differential metabolites enriched in ABC transporters and pyrimidine metabolism were downregulated in the mature milk of sows fed with fermented compound Chinese medicine feed additive. Seven pathways with differential abundances were annotated between colostrum and mature milk of sows with fermented compound Chinese medicine feed additive supplementation (Figure 1C), differential metabolites enriched in purine metabolism were significantly upregulated but in phenylpropanoid biosynthesis were significantly downregulated in colostrum. The differential abundances of metabolic pathways, biosynthesis of secondary metabolites and biosynthesis of various plant secondary metabolites were not significantly different when comparing colostrum to mature milk in sows supplemented without fermented compound Chinese medicine feed additive (Figure 1D); differential metabolites enriched in metabolic pathways, biosynthesis of secondary metabolites and biosynthesis of various plant secondary metabolites were downregulated in colostrum.

The differential abundance score of KEGG metabolic pathways for differential metabolites in milk identified by conventional metabolomics is shown in Figure 2. Differential metabolites of colostrum between experimental and control groups were mapped to three pathways, respectively, and metabolites in the colostrum of the experimental group were significantly upregulated in ABC transporters and purine metabolism, respectively, compared to metabolites in the colostrum of the control group (Figure 2A). Differential metabolites in the mature milk between the experimental and control groups were involved in five pathways and metabolites in the mature milk of the experimental group were significantly upregulated in sucrose metabolism and retrograde endocannabinoid signaling, respectively, compared to those metabolites in the mature milk of the control group (Figure 2B). The differential abundances of 15 metabolic pathways of conventional metabolites between the colostrum and mature milk from the experimental group had significant differences (Figure 2C), differential metabolites involved in purine metabolism, biosynthesis of cofactors, thermogenesis, aldosterone synthesis and secretion, glucagon signaling pathway were upregulated in colostrum, respectively. Fifteen metabolic pathways of differential metabolites between the colostrum and mature milk of the control group had significant differences in differential abundance (Figure 2D), and differential metabolites mapped to the biosynthesis of cofactors were upregulated in the colostrum. 

## 4. Discussion

Milk is the major food source of suckling piglets, and it is one of the crucial factors affecting the survival and growth of suckling piglets [27,28]. Active ingredients in milk are thought to play important roles in the prevention and control of diarrhea during the suckling period [29,30]; however, the level of these active ingredients in milk is generally low and is not enough high for the control of diarrhea, so how to increase the concentration of these active ingredients in milk is an urgent issue to be addressed.

Studies reported that Chinese medicine is a good alternative to in-feed antibiotics because lots of active ingredients in Chinese medicine have antibacterial, anti-inflammatory, antivirus, anti-oxidative and immune-enhancing effects. Our previous study also found that a fermented compound Chinese medicine feed additive had good in vitro effects in inhibiting the growth of *Staphylococcus aureus*, *Salmonella cholerae suis*, *Escherichia coli* and *Streptococcus agalactiae*, because it contained high levels of active ingredients, such as gallic acid, ellagic acid, kaempferide and adenosine [23]. Results of this feeding experiment further showed that the colostrum and mature milk of sows supplemented with fermented compound Chinese medicine feed additive had higher levels of bioactive ingredients than the milk of sows without supplementation of fermented compound Chinese medicine feed additive, particularly, the milk of sows from the experimental group had significantly higher concentrations of quercetin (*p* < 0.05), pinocembrin (*p* < 0.05), chlorogenic acid (*p* < 0.01), methyl succinic acid (*p* < 0.01), L-tryptophan (*p* < 0.01), adenosine (*p* < 0.05), guanine (*p* < 0.05), arteannuin (*p* < 0.05), inosine (*p* < 0.05), guanosine (*p* < 0.05), benzene-1,2,4-triol (*p* < 0.05), hypoxanthine (*p* < 0.05), adenine (*p* < 0.05), ferulic acid (*p* < 0.05), echimidine N-oxide (*p* < 0.05), pogostone (*p* < 0.05), kynurenine (*p* < 0.05) and trehalose 6-phosphate (*p* < 0.05) than the milk of sows from the control group. 

Previous studies evidenced that lots of active ingredients have functions in killing or inhibiting pathogens, alleviating inflammation, enhancing immunity and repairing intestinal barrier function. Quercetin [31], pinocembrin [32,33], pogostone [34], adenosine [35], ferulic acid [36], echimidine-N-oxide [37], purines including adenine, guanine, xanthine and hypoxanthine [38,39,40] have strong antibacterial or antifungal effects on diarrheal pathogens, such as *Escherichia coli*, *Salmonella*, *Staphylococcus aureus*, *Dysentery bacilli*, *Pseudomonas aeruginosa*, *Streptococcus* and *Clostridioides difficile*, in addition, hypoxanthine can speed the excretion of fecal harmful microbiota and toxic substances via shorting gastrointestinal transit time [41]. Pro-inflammatory substances can also cause diarrhea by impairing the intestinal mucosal barrier with pro-inflammatory cytokines [42], but quercetin [31], arteannuin and kynurenine [43,44], ferulic acid [45], purines [46], inosine [47,48], guanosine [49] and benzene-1,2,4-triol [50] can reduce diarrhea by increasing IFN-γ level or inhibiting the production of pro-inflammatory cytokines [48,51,52,53]. Supplementation with Perilla frutescens leaf to Holstein cows changed the composition of differential metabolites in the milk and many differential metabolites with antibacterial and anti-inflammatory effects had been identified [17], results of our experiment also indicated that maternal supplementation with fermented compound Chinese medicine feed additive accumulated lots of high-level differential metabolites in milk which have antibacterial, anti-inflammatory and immune enhancing properties. 

Differential metabolites in colostrum between the experimental group and control group were significantly enriched in the KEGG pathways of flavone and flavanol biosynthesis, galactose metabolism, phenylpropanoid biosynthesis, stilbenoid and gingerol biosynthesis, flavonoid biosynthesis, ABC transporters and purine metabolism, respectively; the colostrum of sows from the experimental group had significantly higher enrichment abundance of differential metabolites than the colostrum of sows from the control group, this means that the colostrum of the experimental group had better antibacterial and anti-inflammatory effects than the colostrum of the control group; differential metabolites in the colostrum of the experimental group can produce other active ingredients in the digestive tract of suckling piglets via these enriched KEGG pathways. It is reported that quercetin can be metabolized into other flavonoids, such as quercitrin, isoquercitrin and myricetin when fermented with some bacteria [54,55]. Adenine, hypoxanthine and guanine can be converted to xanthine and further catabolized to uric acid and allantoin under the action of bacterial fermentation [56,57]. Differential metabolites in mature milk between the experimental group and control group were significantly enriched in the KEGG pathways of starch and sucrose metabolism, and retrograde endocannabinoid signaling; the mature milk of sows from the experimental group had a significantly higher enrichment abundance of differential metabolites than that of sows from the control group. In addition, trehalose-6-phosphate can be mapped onto the pathways of starch and sucrose metabolism and retrograde endocannabinoid signaling. It could be estimated that the mature milk of the experimental group had better functions than the mature milk of the control group in improving the digestion of starter feed and the gut health of suckling piglets; trehalose-6-phosphate can promote the rapid fermentation of carbohydrates, especially glucose and lactose through the pathway of starch and sucrose metabolism to produce volatile fatty acids [58], the rapid fermentation of carbohydrate is particularly important to suckling piglets during the ingestion of starter feed, because mature milk with high trehalose-6-phosphate can quickly lower the intestinal pH of piglets; this is beneficial to the control of gut pathogens and the digestion of nutrients in artificial diets. Retrograde endocannabinoid signaling takes part in the regulation of inflammatory factor release and gut permeability [59], and trehalose-6-phosphate identified in the mature milk of the experimental group is also involved in the pathway of retrograde endocannabinoid signaling, this implies that trehalose-6-phosphate can function in anti-inflammation and intestinal permeability maintenance. Maternal feeding with fermented compound Chinese medicine feed additive had impacts on KEGG pathways and pathway abundances of differential metabolites between colostrum and mature milk. Compared to without supplementation of fermented compound Chinese medicine feed additive, supplementation of fermented compound Chinese medicine feed additive to sows from d 104 of gestation to d 25 of lactation elevated the number of enriched KEGG pathways; increased the enrichment abundance of purine metabolism, the glucagon signaling pathway, aldosterone synthesis and secretion, and the thermogenesis of differential metabolites in colostrum metabolism, but decreased the enrichment abundance of protein digestion and absorption, biosynthesis of amino acid, D-amino acid metabolism and purine metabolism of differential metabolites in mature milk metabolism. Increasing the enrichment abundance of glucagon signaling and the thermogenesis pathways could metabolize nutrients to produce more heat to raise the cold resistance of newborn animals [60,61], it is very important for newborn piglets, because the increased cold resistance can decrease the morbidity of newborn piglets.

## 5. Conclusions

Maternal feeding with fermented Chinese medicine feed additive elevated the concentrations of functional ingredients in the colostrum and mature milk of sows, and most of these functional ingredients have anti-infectious, anti-inflammatory, anti-oxidative and immune-enhancing effects. 

## Figures and Tables

**Figure 1 animals-12-02879-f001:**
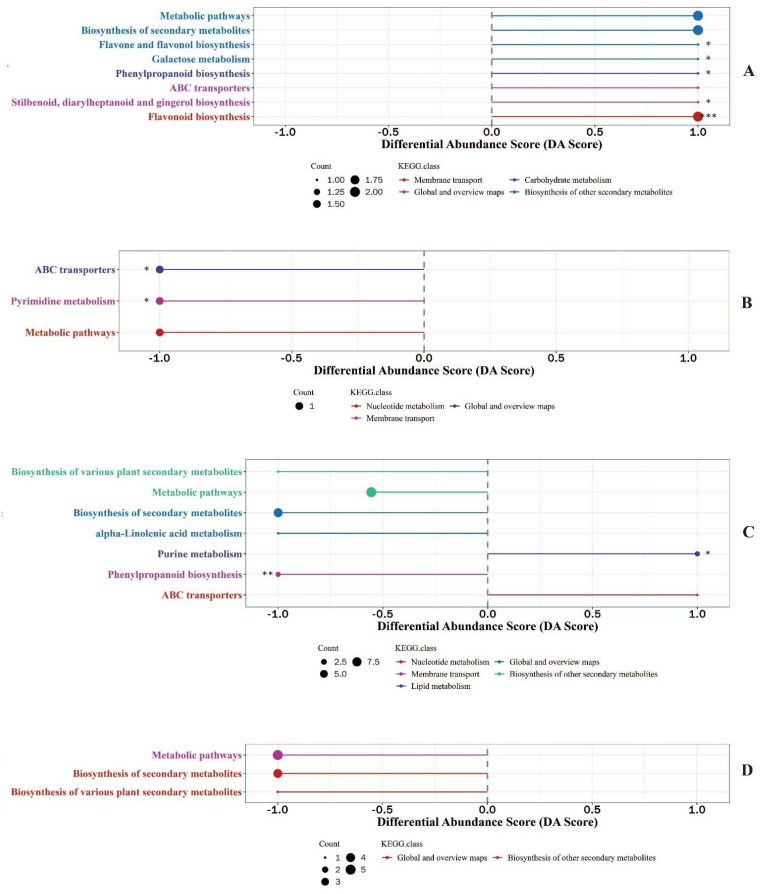
Differential abundance score of metabolic pathways of metabolites identified in milk using chinmedomics approach. (**A**): KEGG pathways enrichment of differential metabolites in colostrum between experimental group and control group. (**B**): KEGG pathways enrichment of differential metabolites in mature milk between experimental group and control group. (**C**): KEGG pathways enrichment of differential metabolites between colostrum and mature milk of experimental group. (**D**): KEGG pathways enrichment of differential metabolites between colostrum and mature milk of control group. * 0.01 < *p* <0.05, ** 0.001 < *p* <0.01, *** *p* < 0.001.

**Figure 2 animals-12-02879-f002:**
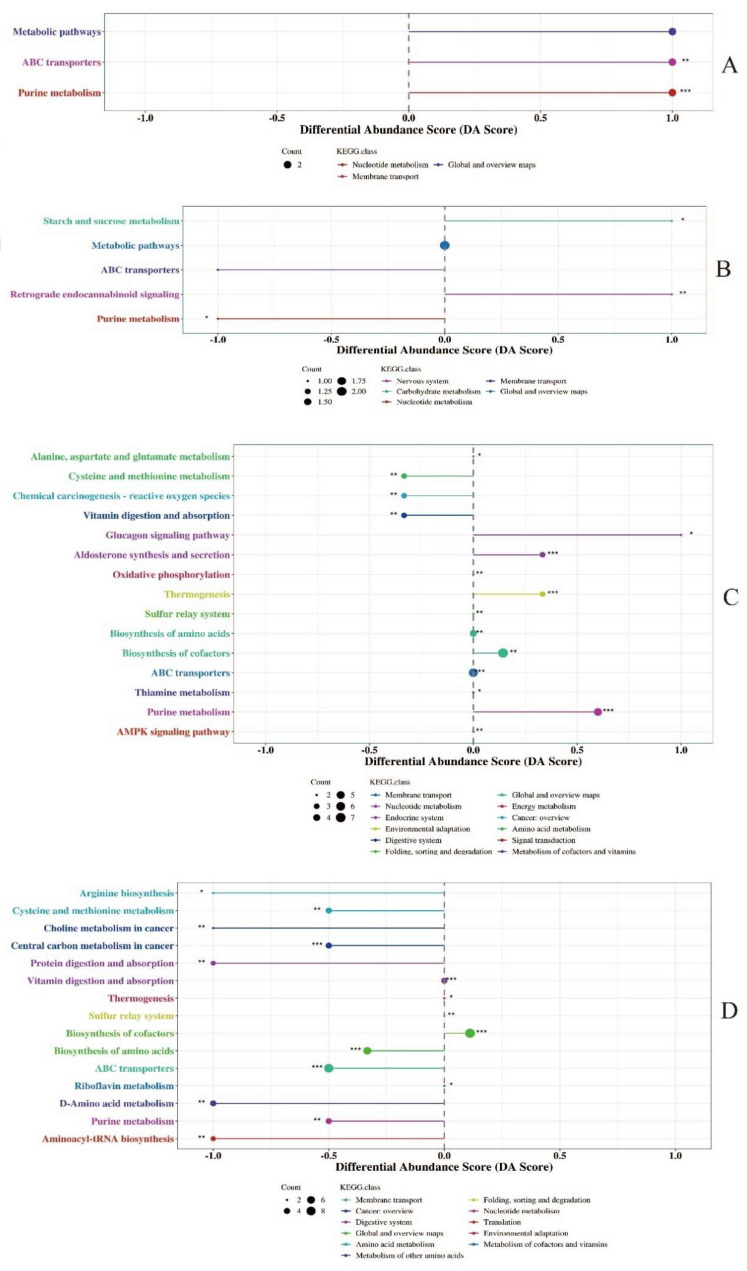
Differential abundance score of metabolic pathways of metabolites identified in milk using conventional metabolomics approach. (**A**): KEGG pathways enrichment of differential metabolites in colostrum between experimental group and control group. (**B**): KEGG pathways enrichment of differential metabolites in mature milk between experimental group and control group. (**C**): KEGG pathways enrichment of differential metabolites between colostrum and mature milk of experimental group. (**D**): KEGG pathways enrichment of differential metabolites between colostrum and mature milk of control group. * 0.01 < *p* < 0.05, ** 0.001 < *p* <0.01, *** *p* < 0.001.

**Table 1 animals-12-02879-t001:** Composition and nutrient levels of the basal diet (air-dry basis) %.

Items	Content
Ingredients	
Corn	60.00
Wheat bran	15.00
Soybean meal	15.00
Rapeseed meal	4.00
Fish meal	2.00
Premix ^a^	4.00
Total	100.00
Nutrient levels ^b^	
Metabolizable energy (MJ/kg)	11.91
Crude protein	17.35
Ether extract	4.73
Crude fiber	4.12
Calcium	0.83
Total phosphorus	0.65
Lysine	0.94
Methionine + Cystine	0.72
Threonine	0.53

^a^ Per kg of premix provided the following: VA 421,000 IU, VD 61,000 IU, VE 940 IU, VK_3_ 125 mg, VB_1_ 65 mg, VB_2_ 230 mg, VB_6_ 75 mg, VB_12_ 1.0 mg, nicotinic acid 1 100 mg, *D*-pantothenic acid 680 mg, folic acid 45 mg, choline 12 g, Fe 1.5 g, Cu 0.8 g, Zn 1.6 g, Mn 85 g, I 8 mg, Co 30 mg, Se 7 mg, Ca 185 mg, P 18 mg, NaCl 100,000 mg, ^b^ Levels of metabolizable energy and amino acids were calculated values, while the others were measured values.

**Table 2 animals-12-02879-t002:** Differential metabolites in colostrum between experimental and control groups based on untargeted chinmedomics.

MS2 Name	MS2 Score	Average of Differential Metabolites in Colostrum Based on Chinmedomics(×10^−3^)		VIP	*p*-Value	Fold Change (A/B)
		Experimental Group (A)	Control Group (B)			
**Negative ion model**						
** *Flavonoids* **						
Wogonin	0.996	2.744	0.035	1.322	0.363	78.400
Quercetin-3-O-galactoside	0.969	0.286	0.035	1.552	0.232	8.171
Quercetin	0.854	0.257	0.035	3.444	0.039	7.343
Isorhamnetin	0.999	0.416	0.063	1.058	0.212	6.603
Mulberrin	0.843	0.203	0.035	1.868	0.367	5.800
Pinocembrin	0.855	2.284	0.034	2.188	0.033	67.176
Daidzein	0.986	19.279	9.152	1.137	0.518	2.106
Corylin	0.901	0.750	0.539	2.012	0.627	1.392
** *Phenols* **						
4-Nitrophenol	0.827	0.097	0.035	1.778	0.374	2.771
P-Anisic acid	0.910	5.020	0.741	1.540	0.266	6.775
6-Gingerol	0.933	837.143	643.369	1.927	0.349	1.301
** *Alkaloids* **						
Thymidine	0.865	27.459	26.151	1.853	0.872	1.050
* **Phenylpropanoids** *						
Chlorogenic acid	0.834	0.490	0.034	2.532	0.008	14.412
Methyl chlorogenate	0.835	8.172	6.547	1.483	0.355	1.248
** *Terpenoids* **						
alpha-Hederin	0.963	0.347	0.035	1.612	0.183	9.914
** *Fatty Acyls* **						
Citraconic acid	0.982	47.162	0.035	1.321	0.363	1347.486
Methyl hexadecanoate	0.987	74.305	19.187	1.832	0.359	3.873
** *Fatty acids* **						
Methyl succinic acid	0.958	21.930	0.033	2.686	0.008	664.545
Pimelic acid	0.809	11.711	6.641	1.799	0.270	1.763
** *Organoheterocyclic compounds* **						
L-Tryptophan	0.851	578.724	232.278	1.135	0.002	2.492
** *Organic acids and derivatives* **						
Threonic acid	0.979	30.264	17.162	1.525	0.647	1.763
Citrate	0.998	5601.115	5086.045	1.883	0.900	1.101
** *Organic oxygen compounds* **						
Melibiose	1.000	133.084	0.035	2.558	0.099	3802.400
** *Amino acid derivatives* **						
Aspartate	0.890	18.378	0.035	2.036	0.274	525.086
** *Carbohydrates and derivatives* **						
Glyceric acid	0.984	3.274	0.035	1.975	0.363	93.543
** *Aliphatics* **						
Octyl gallate	0.888	0.527	0.035	1.278	0.361	15.057
**Positive ion model**						
** *Alkaloids* **						
Adenosine	0.883	36.193	6.189	1.090	0.015	5.848
L-Carnitine	0.959	45.650	23.009	2.491	0.236	1.984
Guanine	1.000	111.347	33.426	2.084	0.031	3.331
Guanosine	0.997	49.249	32.999	2.086	0.208	1.492
Trigonelline HCl	0.996	3.130	2.334	1.578	0.275	1.341
Jatrorrhizine	0.941	2.599	2.361	1.004	0.508	1.101
Palmatine	0.984	8.683	8.067	1.502	0.231	1.076
** *Phenols* **						
Bergenin	0.936	2.310	1.654	1.296	0.429	1.397
Gallic acid	0.908	15.722	14.697	1.066	0.946	1.070
** *Terpenoids* **						
Resibufogenin	0.997	1.661	0.003	1.402	0.363	553.667
Estradiol	0.835	0.121	0.003	1.387	0.364	40.333
Lovastatin	0.993	0.560	0.444	1.285	0.593	1.261
Neoandrogsinapixrapholide	0.947	4.384	3.605	2.065	0.153	1.216
Lindenenol	0.834	1.834	1.583	2.209	0.100	1.159
Curcumenol	0.870	6.618	5.974	1.899	0.153	1.108
Arteannuin	0.956	1.103	0.941	2.778	0.049	1.172
* **Organooxygen compounds** *						
Pogostone	0.808	1.057	0.013	3.245	0.242	81.308
** *Amino acid derivatives* **						
Isoleucine	0.881	3.730	0.003	1.404	0.363	1243.333
Aspartic acid	0.957	0.596	0.003	2.313	0.178	198.667
Lysine	0.856	1.507	0.130	1.270	0.216	11.592
L-Isoleucine	0.953	913.081	209.378	1.402	0.315	4.361
Arginine	0.941	5.951	2.321	1.134	0.328	2.564

**Table 3 animals-12-02879-t003:** Differential metabolites in mature milk between experimental and control groups based on untargeted chinmedomics.

MS2 Name	MS2 Score	Average of Differential Metabolites in Mature milk Based on Chinmedomics (×10^−3^)		VIP	*p*-Value	Fold Change (C/D)
		Experimental group (C)	Control group (D)			
**Negative ion model**						
** *Phenols* **						
Gallic acid	0.990	3.626	0.043	1.643	0.349	84.326
4-hydroxybenzaldehyde	1.000	1.673	0.043	1.056	0.363	38.907
Methyl gallate	0.993	1.087	0.043	1.202	0.363	25.280
Orcinol	0.928	0.212	0.043	1.208	0.362	4.930
** *Phenylpropanoids* **						
Danshensu	0.812	0.991	0.043	1.202	0.363	23.047
rosmarinic acid	1.000	0.318	0.043	1.206	0.362	7.395
Sinapic acid	0.979	0.193	0.043	1.945	0.164	4.488
Ferulic acid	0.977	6.375	2.057	2.092	0.020	3.099
** *Xanthones* **						
Gentisic acid	0.840	1.230	0.043	1.201	0.363	28.605
** *Sesquiterpenoids* **						
Abscisic acid	0.972	0.283	0.043	1.081	0.360	6.581
** *Amino acid derivatives* **						
N-Acetyl-DL-glutamic acid	0.908	0.436	0.043	1.205	0.363	10.140
**Positive ion model**						
** *Alkaloids* **						
Boldine	0.843	0.124	0.003	1.220	0.363	41.333
Securinine	0.824	0.037	0.003	1.218	0.363	12.333
Actidione	0.900	0.022	0.005	2.188	0.072	4.400
Echimidine N-oxide	0.835	1.010	0.758	2.126	0.026	1.333
** *Flavonoids* **						
Kaempferol	0.995	0.007	0.003	1.106	0.350	2.333
Isoquercitrin	0.822	0.006	0.003	1.001	0.342	2.000
Biochanin A	0.977	3.386	2.587	1.187	0.263	1.309
** *Chalcones* **						
Loureirin A	0.952	0.017	0.003	1.215	0.364	5.667
** *Phenylpropanoids* **						
Ethyl ferulate	0.963	0.070	0.003	1.237	0.363	23.333
p-Coumaric acid	0.976	0.137	0.039	1.137	0.133	3.513
Eudesmin	0.837	0.970	0.451	1.354	0.552	2.151
** *Terpenoids* **						
Grosheimin	0.833	0.017	0.003	1.254	0.360	5.667
Reynosin	0.855	0.012	0.003	1.078	0.358	4.000
Estradiol	0.835	0.312	0.137	1.068	0.593	2.277
Andrographolide	0.932	0.025	0.014	1.359	0.307	1.786
** *Sesquiterpenoids* **						
Germacrone	0.947	3.949	2.849	1.241	0.287	1.386
** *Organooxygen compounds* **						
Pogostone	0.807	0.487	0.003	3.311	0.021	162.333
** *Fatty acids* **						
Chaulmoogric Acid	0.920	1.686	1.351	1.856	0.549	1.248
** *Amino acid derivatives* **						
Isoleucine	0.881	1.588	0.003	1.222	0.363	529.333
Kynurenine	0.928	1.038	0.433	2.093	0.012	2.397
** *Carboxylic acids and derivatives* **						
L-Tyrosine	0.914	1.171	0.003	1.384	0.360	390.333

**Table 4 animals-12-02879-t004:** Differential metabolites between colostrum and mature milk of experimental group based on untargeted chinmedomics.

MS2 Name	MS2 Score	Average of Differential Metabolites in Experimental Group Based on Chinmedomics (×10^−3^)		VIP	*p*-Value	Fold Change (A/C)
		Colostrum (A)	Mature Milk (C)			
**Negative ion model**						
** *Flavonoids* **						
Pinocembrin	0.855	1.163	0.045	1.076	0.091	25.844
**Positive ion model**						
** *Phenols* **						
Bergenin	0.936	2.310	0.003	2.621	0.027	770.000
** *Alkaloids* **						
3-Furfuryl 2-pyrrolecarboxylate	0.887	0.806	0.003	2.364	0.010	268.667
Boldine	0.843	11.892	0.124	2.554	0.052	95.903
Adenosine	0.883	60.919	8.436	1.804	0.215	7.221
Guanosine	0.997	49.249	12.557	2.397	0.008	3.922
Guanine	1.000	0.083	0.0252	2.281	0.017	3.290
Nicotinamide	0.920	75.339	26.165	1.407	0.321	2.879
Jatrorrhizine	0.941	2.599	2.011	1.220	0.137	1.292
Palmatine	0.984	8.683	6.861	2.029	0.006	1.266
** *Terpenoids* **						
Beta-Caryophyllene alcohol	0.872	46.866	0.092	2.628	0.080	509.413
Cortodoxone	0.901	0.210	0.003	1.013	0.364	70.000
Celastrol	0.864	7.865	2.272	1.953	0.016	3.462
Lindenenol	0.834	1.834	1.481	1.758	0.027	1.238
Artemisinin	0.873	2.594	2.120	1.906	0.012	1.224
Curcumenol	0.870	6.618	5.482	1.740	0.033	1.207
Aucubin	0.882	0.232	0.003	2.652	0.020	77.333
** *Phenylpropanoids* **						
Suberosin	0.898	0.878	0.456	1.947	0.051	1.925
Eudesmin	0.837	1.768	0.970	1.729	0.412	1.8230
** *Coumarins and derivatives* **						
7-Hydroxycoumarin	0.909	1.183	0.841	1.026	0.139	1.407
* **Phospholipids** *						
Monolinolein	0.957	0.548	0.441	1.002	0.761	1.243
** *Organoheterocyclic compounds* **						
L-Tryptophan	0.846	0.245	0.086	1.060	0.278	2.846
** *Fatty acids* **						
Chaulmoogric acid	0.920	2.006	1.686	1.376	0.832	1.190

**Table 5 animals-12-02879-t005:** Differential metabolites between colostrum and mature milk of control group based on untargeted chinmedomics.

MS2 Name	MS2 Score	Average of Differential Metabolites in Control Group Based on Chinmedomics (×10^−3^)		VIP	*p*-Value	Fold Change (B/D)
		Colostrum (B)	Mature Milk (D)			
**Negative Ion Model**						
** *Organic acids and derivatives* **						
Threonic acid	0.979	17.162	5.260	1.041	0.016	3.263
**Positive ion model**						
** *Phenols* **						
Bergenin	0.936	1.654	0.003	3.086	0.002	551.333
** *Alkaloids* **						
Boldine	0.843	7.322	0.003	3.083	0.016	2440.667
3-Furfuryl 2-pyrrolecarboxylate	0.887	0.761	0.003	2.800	0.013	253.667
L-Phenylalanine	0.980	2.546	0.364	1.655	0.137	6.995
Nicotinamide	0.920	58.747	23.622	1.372	0.298	2.487
Adenosine	0.883	19.129	8.003	1.088	0.200	2.390
Guanosine	0.997	32.999	19.168	1.448	0.158	1.722
Guanine	1.000	54.502	32.312	1.431	0.132	1.687
Echimidine N-oxide	0.835	1.046	0.758	1.785	0.074	1.380
** *Terpenoids* **						
Beta-Caryophyllene alcohol	0.872	35.370	0.003	2.340	0.243	116.733
Judaicin	0.943	0.304	0.003	1.062	0.363	101.333
Cortodoxone	0.901	0.065	0.003	1.047	0.363	21.667
Dehydrocostus lactone	0.977	0.026	0.003	1.045	0.364	8.667
Aucubin	0.882	0.360	0.003	2.476	0.033	120.000
Celastrol	0.864	8.732	3.345	2.352	0.002	2.610
** *Phenylpropanoids* **						
Eudesmin	0.837	2.262	0.451	2.562	0.001	5.016
Suberosin	0.898	0.790	0.587	1.506	0.190	1.346
** *Coumarins and derivatives* **						
7-Hydroxycoumarin	0.909	1.047	0.743	1.365	0.267	1.409
** *Organooxygen compounds* **						
Pogostone	0.808	0.013	0.003	1.043	0.364	4.333
** *Benzene and substituted derivatives* **						
Phenethylacetate	0.868	0.022	0.003	1.651	0.182	7.333
** *Amino acid derivatives* **						
Kynurenine	0.928	1.325	0.433	1.491	0.246	3.060

**Table 6 animals-12-02879-t006:** Differential metabolites in colostrum between experimental and control groups based on conventional untargeted metabolomics.

MS2 Name	MS2 Score	Average of Differential Metabolites in Colostrum Based on Conventional Untargeted Metabolomics		VIP	*p*-Value	Fold Change (A/B)
		Experimental Group (A)	Control Group (B)			
**Negative ion model**						
** *Organic acids and derivatives* **						
L-Phenylalanine	0.959	1.712	0.393	1.001	0.252	4.356
Succinic acid	0.965	0.710	0.301	1.220	0.322	2.359
Maleic acid	0.995	0.518	0.372	1.203	0.431	1.392
Citric acid	0.989	94.808	71.197	1.490	0.118	1.332
Glycine	0.994	0.043	0.032	1.295	0.203	1.344
Malonic acid	0.957	2.430	2.094	1.069	0.319	1.160
Pyruvic acid	0.987	3.497	3.047	1.106	0.466	1.148
Creatinine	0.805	2.702	2.420	1.145	0.556	1.117
Acrylic acid	0.994	1.519	1.386	1.057	0.466	1.096
** *Organoheterocyclic compounds* **						
2-Hydroxyxanthone	0.970	0.035	0.029	1.543	0.144	1.207
Quinolinic acid	0.913	0.493	0.404	1.039	0.212	1.220
Pyrrole-2-carboxylic acid	0.997	0.764	0.636	1.151	0.452	1.201
Guanine	0.985	0.846	0.358	1.264	0.163	2.363
** *Organooxygen compounds* **						
D-Ribulose 5-phosphate	0.903	1.045	0.814	1.116	0.242	1.284
** *Organic oxygen compounds* **						
N-Acetylneuraminic acid	0.988	0.762	0.409	1.492	0.209	1.863
Myo-Inositol	0.911	20.558	16.271	1.208	0.229	1.263
6-Phosphogluconic acid	0.841	0.547	0.242	1.328	0.136	2.260
** *Nucleosides, nucleotides, and analogues* **						
Uridine diphosphate glucuronic acid	0.842	2.257	1.036	1.209	0.258	2.179
Uridine diphosphategalactose	0.882	4.969	2.321	1.372	0.107	2.141
Uridine 5′-diphosphate	0.903	0.423	0.204	1.575	0.158	2.074
Inosine	0.962	2.588	1.361	2.424	0.017	1.902
Guanosine	0.944	1.024	0.581	1.912	0.043	1.762
Uridine 5′-monophosphate	0.867	32.048	20.522	1.047	0.260	1.562
2-Methylguanosine	0.865	0.034	0.024	1.654	0.143	1.417
S-Adenosylhomocysteine	0.836	0.037	0.027	1.951	0.121	1.370
** *Lipids and lipid-like molecules* **						
Tetradecanedioic acid	0.977	0.019	0.013	1.298	0.216	1.462
9-Decenoic acid	0.998	0.031	0.018	1.991	0.104	1.722
12-Methyltridecanoic acid	0.985	0.507	0.294	1.562	0.139	1.724
Heptanoic acid	0.856	0.171	0.134	1.638	0.137	1.276
FA(18:2)	1.000	0.817	0.216	1.042	0.373	3.782
LPC(16:0)	0.891	0.390	0.106	1.086	0.267	3.679
LPC(18:1)	0.850	0.234	0.082	1.829	0.180	2.854
Dodecanedioic acid	0.912	0.025	0.012	1.579	0.158	2.083
** *Benzenoids* **						
benzene-1,2,4-triol	0.928	0.362	0.219	2.241	0.037	1.653
Butylparaben	0.979	0.108	0.086	1.550	0.125	1.256
4-Nitrophenol	1.000	0.410	0.343	1.811	0.105	1.195
N-acetyl-5-aminosalicylic acid	0.891	0.076	0.024	1.423	0.123	3.167
**Positive ion model**						
** *Organic acids and derivatives* **						
Phenylalanylproline	0.969	0.112	0.011	1.460	0.186	10.182
ACar(18:1)	0.883	0.684	0.075	1.227	0.348	9.120
ACar(14:0)	0.959	0.129	0.014	1.282	0.331	9.214
ACar(6:1)	0.997	0.158	0.052	1.918	0.091	3.038
ACar(6:0)	0.993	1.050	0.641	1.267	0.224	1.638
L-Glutamine	0.995	0.108	0.014	1.389	0.247	7.714
1-Methylhistidine	0.943	0.091	0.014	1.463	0.241	6.500
ACar(16:1)	0.960	0.087	0.014	1.402	0.311	6.214
ACar(8:0)	0.979	0.032	0.006	2.478	0.091	5.333
L-Tryptophan	0.978	0.377	0.089	1.015	0.285	4.236
Pipecolic acid	0.983	0.180	0.172	1.018	0.523	1.047
Elenaic acid	0.871	0.042	0.022	1.180	0.198	1.909
Proline betaine	0.941	0.301	0.240	1.187	0.442	1.254
Betaine	0.999	2.490	2.144	1.066	0.573	1.161
Palmitoylethanolamide	0.897	0.309	0.105	1.342	0.259	2.943
** *Organic nitrogen compounds* **						
L-Carnitine	0.996	0.405	0.231	1.869	0.178	1.753
** *Organic oxygen compounds* **						
Adenosine 2’-phosphate	0.943	1.002	0.444	1.170	0.448	2.257
** *Organoheterocyclic compounds* **						
Pyridoxal	0.996	0.143	0.117	1.432	0.185	1.222
Hypoxanthine	1.000	1.426	0.907	2.027	0.035	1.572
3-Pyridinebutanoic acid	0.973	1.416	0.331	1.324	0.267	4.278
Adenine	0.998	0.081	0.021	1.825	0.017	3.857
** *Nucleosides, nucleotides, and analogues* **						
5′-Methylthioadenosine	0.988	0.023	0.010	1.006	0.245	2.300
Guanosine diphosphate	0.976	0.575	0.266	1.732	0.157	2.162
Deoxyguanosine	1.000	0.098	0.059	1.534	0.191	1.661
** *Lipids and lipid-like molecules* **						
L-Palmitoylcarnitine	0.957	1.813	0.121	1.120	0.344	14.983
LysoPE(20:1(11Z)/0:0)	0.931	0.019	0.005	1.382	0.225	3.800
LysoPE(18:1(11Z)/0:0)	0.898	1.032	0.539	1.981	0.084	1.915
LysoPE(16:1(9Z)/0:0)	0.970	0.025	0.010	1.359	0.245	2.500
PE(14:1(9Z)/14:0)	0.912	0.025	0.009	1.508	0.172	2.778
LPC(16:1)	0.888	0.125	0.056	1.145	0.340	2.232
LPC(18:0)	0.879	0.145	0.069	1.790	0.210	2.101
LPE(18:0)	0.891	2.616	1.251	1.736	0.130	2.091
Oleamide	0.992	0.814	0.396	1.027	0.278	2.056
Glycerol tripropanoate	0.999	0.034	0.019	1.422	0.155	1.789
Cohibin C	0.831	0.218	0.189	1.116	0.446	1.153
Stearoylcarnitine	0.925	0.598	0.082	1.456	0.276	7.293
L-Acetylcarnitine	0.929	14.567	11.410	1.109	0.337	1.277
** *Benzenoids* **						
Dibutyl phthalate	0.998	0.094	0.089	1.288	0.287	1.056
p-Aminobenzoic acid	0.983	0.126	0.051	1.237	0.367	2.471

**Table 7 animals-12-02879-t007:** Differential metabolites in mature milk between experimental and control groups based on conventional untargeted metabolomics.

MS2 Name	MS2 Score	Average of Differential Metabolites in Mature Milk Based on Conventional Untargeted Metabolomics		VIP	*p*-Value	Fold Change (C/D)
		Experimental Group (C)	Control Group (D)			
**Negative ion model**						
** *Organic acids and derivatives* **						
Glycolic acid	1.000	2.025	1.768	1.167	0.322	1.145
3-Sialyl-N-acetyllactosamine	0.895	0.081	0.016	1.930	0.337	5.063
N-Acetylneuraminic acid	0.988	0.047	0.021	2.355	0.096	2.238
Trehalose 6-phosphate	0.948	0.552	0.345	2.377	0.047	1.600
Gluconic acid	0.934	1.145	0.893	1.009	0.587	1.282
** *Organoheterocyclic compounds* **						
Pyrrole-2-carboxylic acid	0.997	1.081	0.782	1.790	0.221	1.382
** *Nucleosides, nucleotides, and analogues* **						
Uridine 5′-monophosphate	0.867	14.577	7.388	1.206	0.275	1.973
S-Adenosylhomocysteine	0.836	0.015	0.010	1.094	0.490	1.500
** *Lipids and lipid-like molecules* **						
SHexCer(d30:3)	1.000	0.736	0.017	1.445	0.368	43.294
FA(20:5)	1.000	0.039	0.017	2.205	0.165	2.294
Dihydrojasmonic acid	0.968	0.155	0.121	1.232	0.458	1.281
** *Ben* ** *zenoids*						
4-Nitrophenol	1.000	0.520	0.447	1.642	0.311	1.163
benzene-1,2,4-triol	0.928	0.469	0.363	1.938	0.097	1.292
**Positive ion model**						
** *Organic acids and derivatives* **						
O-Acetylserine	0.990	0.061	0.046	1.781	0.145	1.326
Oxypinnatanine	0.995	0.125	0.112	1.641	0.255	1.116
** *Organic oxygen compounds* **						
Falcarinone	0.992	0.231	0.177	1.066	0.283	1.305
N,O-Didesmethylvenlafaxine	0.997	0.486	0.441	1.826	0.122	1.102
L-Gulose	0.986	0.436	0.388	1.128	0.694	1.124
** *Organic nitrogen compounds* **						
Choline	1.000	1.134	0.820	2.212	0.110	1.383
** *Organoheterocyclic compounds* **						
5-Methyl-2(3H)-furanone	0.956	0.316	0.303	1.078	0.621	1.043
** *Lipids and lipid-like molecules* **						
Montecristin	0.837	0.031	0.024	1.593	0.426	1.292
Ginkgolide J	0.958	0.039	0.032	1.295	0.501	1.219
** *Nucleosides, nucleotides, and analogues* **						
Guanosine	0.997	0.151	0.139	1.099	0.719	1.086

**Table 8 animals-12-02879-t008:** Differential metabolites between colostrum and mature milk of experimental group based on conventional untargeted metabolomics.

MS2 Name	MS2 Score	Average of Differential Metabolites in Experimental Group Based on Conventional Untargeted Metabolomics		VIP	*p*-Value	Fold Change (A/C)
		Colostrum (A)	Mature Milk (C)			
**Negative ion model**						
** *Organic acids and derivatives* **						
Citric acid	0.989	94.808	47.751	1.341	0.010	1.985
D-Alanine	0.999	2.926	2.086	1.162	0.024	1.403
N-Acetylneuraminic acid	0.988	0.762	0.047	1.787	0.031	16.213
6-Phosphogluconic acid	0.841	0.547	0.035	1.326	0.028	15.629
3′-Sialyllactose	0.908	13.631	1.917	1.695	0.008	7.111
3-Sialyl-N-acetyllactosamine	0.895	0.532	0.081	1.433	0.044	6.568
Gluconic acid	0.934	2.801	1.145	1.339	0.030	2.446
** *Organoheterocyclic compounds* **						
Riboflavin	0.968	0.170	0.015	1.619	0.072	11.333
** *Lipids and lipid-like molecules* **						
PI(18:2/18:2)	0.873	2.903	2.097	1.027	0.138	1.384
PI(18:1/20:3)	0.876	23.702	3.781	1.713	0.000	6.269
PI(18:0/20:3)	0.889	4.429	1.115	1.696	0.001	3.972
PI(18:0/22:6)	0.825	0.209	0.063	1.551	0.043	3.317
PI(18:0/18:1)	0.915	2.468	1.189	1.269	0.085	2.076
PI(18:0/18:2)	0.886	25.133	12.184	1.307	0.078	2.063
OxPI(18:0/18:1 + 3O)	0.957	0.488	0.094	1.732	0.006	5.191
OxPI(16:0/18:1 + 3O)	0.954	4.880	3.619	1.036	0.119	1.348
FA(19:4)	1.000	0.057	0.006	1.113	0.313	9.500
FA(16:2)	1.000	20.175	0.894	1.057	0.315	22.567
** *Nucleosides, nucleotides, and analogues* **						
2-Methylguanosine	0.865	0.034	0.018	1.202	0.021	1.889
**Positive ion model**						
** *Organic acids and derivatives* **						
Elenaic acid	0.871	0.042	0.007	1.713	0.043	6.000
Palmitoylethanolamide	0.897	0.309	0.084	1.089	0.218	3.679
ACar(6:1)	0.997	0.158	0.015	1.697	0.037	10.533
trans-Aconitic acid	0.803	0.193	0.052	1.758	0.004	3.712
** *Organic oxygen compounds* **						
Picraquassioside A	0.994	0.158	0.009	1.888	0.039	17.556
Pseudouridine 5′-phosphate	0.921	3.103	0.661	1.298	0.027	4.694
L-Gulose	0.986	1.744	0.436	1.877	0.014	4.000
Falcarinone	0.992	0.893	0.231	1.626	0.043	3.866
N-Acetylmannosamine	0.927	0.388	0.101	1.603	0.035	3.842
** *Organoheterocyclic compounds* **						
Isolinderanolide	0.854	0.063	0.016	1.052	0.303	3.938
Guanine	0.999	0.933	0.319	1.243	0.122	2.925
Hypoxanthine	1.000	1.426	0.711	1.468	0.006	2.006
Thiamine	0.997	0.690	0.187	1.442	0.038	3.690
Safrole	0.974	3.080	2.859	1.284	0.022	1.077
** *Organohalogen compounds* **						
Chloral hydrate	0.907	0.087	0.018	1.736	0.023	4.833
** *Lipids and lipid-like molecules* **						
13-HOTE	0.813	0.292	0.010	1.129	0.307	29.200
Caryoptosidic acid	0.833	0.014	0.001	1.911	0.012	14.000
Stearoylcarnitine	0.925	0.598	0.047	1.220	0.248	12.723
LPC(22:4)	0.869	0.029	0.003	1.384	0.286	9.667
LPC(22:5)	0.834	0.028	0.005	1.684	0.122	5.600
L-Acetylcarnitine	0.929	14.567	3.852	1.803	0.012	3.782
Asitrilobin C	0.813	0.152	0.069	1.687	0.001	2.203
Turanose	1.000	0.849	0.777	1.062	0.913	1.093
** *Nucleosides, nucleotides, and analogues* **						
Cyclic AMP	0.897	0.041	0.002	1.220	0.066	20.500
Cyclic GMP	0.924	0.194	0.023	1.671	0.020	8.435
N6-Methyladenosine	0.999	0.730	0.158	1.549	0.020	4.620
Guanosine	0.997	0.541	0.133	1.787	0.002	4.068
S-Adenosylhomocysteine	0.861	0.045	0.012	1.592	0.000	3.750
1-Methylguanosine	0.992	0.051	0.023	1.717	0.001	2.217
Inosine	0.999	0.615	0.289	1.505	0.007	2.128
7-Methylinosine	0.960	0.014	0.007	1.556	0.006	2.000
** *Benzenoids* **						
N-cis-Feruloyltyramine	0.992	0.024	0.014	1.258	0.041	1.714

**Table 9 animals-12-02879-t009:** Differential metabolites between colostrum and mature milk of control group based on conventional untargeted metabolomics.

MS2 Name	MS2 Score	Average of Differential Metabolites in Control Group Based on Conventional Untargeted Metabolomics		VIP	*p*-Value	Fold Change (B/D)
		Colostrum (B)	Mature Milk (D)			
**Negative ion model**						
** *Organic acids and derivatives* **						
Indoxyl sulfate	0.982	0.139	0.084	1.078	0.339	1.655
Citric acid	0.989	71.197	47.904	1.024	0.014	1.486
** *Organic oxygen compounds* **						
3-Sialyl-N-acetyllactosamine	0.895	0.520	0.016	1.481	0.009	32.500
N-Acetylneuraminic acid	0.988	0.409	0.021	1.578	0.002	19.476
6-Phosphogluconic acid	0.841	0.242	0.043	1.323	0.023	5.628
Trehalose 6-phosphate	0.948	1.681	0.345	1.268	0.048	4.872
Gluconic acid	0.934	2.744	0.893	1.205	0.003	3.073
** *Organoheterocyclic compounds* **						
Dehydroascorbic acid	0.963	5.568	2.982	1.298	0.004	1.867
** *Lipids and lipid-like molecules* **						
PI(18:1/20:3)	0.876	22.868	4.606	1.479	0.001	4.965
PI(18:0/22:6)	0.825	0.240	0.081	1.365	0.021	2.963
PI(18:0/20:3)	0.889	4.412	1.285	1.422	0.008	3.433
PI(18:0/18:2)	0.886	24.813	13.422	1.169	0.051	1.849
**Positive ion model**						
** *Organic acids and derivatives* **						
ACar(6:1)	0.997	0.052	0.023	1.125	0.055	2.261
ACar(5:0)	1.000	41.315	22.319	1.147	0.067	1.851
Elenaic acid	0.871	0.022	0.008	1.113	0.103	2.750
** *Organic oxygen compounds* **						
Picraquassioside A	0.994	0.122	0.006	1.691	0.008	20.333
Pseudouridine 5′-phosphate	0.921	2.390	0.176	1.501	0.021	13.580
Falcarinone	0.992	1.017	0.177	1.818	0.000	5.746
L-Gulose	0.986	2.105	0.388	1.065	0.000	5.425
N-Acetylmannosamine	0.927	0.434	0.093	1.794	0.003	4.667
3′-Sialyllactose	0.924	0.381	0.035	1.746	0.002	10.886
2-Carboxyarabinitol 5-phosphate	0.888	0.152	0.070	1.293	0.008	2.171
** *Organoheterocyclic compounds* **						
Riboflavin	0.995	0.851	0.036	1.681	0.012	23.639
Guanine	0.999	0.540	0.268	1.235	0.058	2.015
Thiamine	0.997	0.467	0.240	1.113	0.047	1.946
** *Organohalogen compounds* **						
Chloral hydrate	0.907	0.105	0.015	1.608	0.001	7.000
** *Nucleosides, nucleotides, and analogues* **						
Cyclic AMP	0.897	0.030	0.001	1.074	0.095	30.000
Cyclic GMP	0.924	0.179	0.026	1.655	0.019	6.885
S-Adenosylhomocysteine	0.861	0.037	0.009	1.717	0.000	4.111
N6-Methyladenosine	0.999	0.551	0.148	1.531	0.015	3.723
Guanosine monophosphate	0.993	0.133	0.038	1.217	0.077	3.500
7-Methylinosine	0.960	0.018	0.009	1.722	0.000	2.000
1-Methylguanosine	0.992	0.049	0.026	1.822	0.000	1.885
Guanosine	0.997	0.320	0.209	1.005	0.252	1.531
** *Lipids and lipid-like molecules* **						
LPC(22:4)	0.869	0.008	0.002	1.151	0.079	4.000
L-Acetylcarnitine	0.929	11.410	3.590	1.684	0.001	3.178
Asitrilobin C	0.813	0.154	0.069	1.563	0.001	2.232

## Data Availability

The data presented in this study are available in [Untargeted Metabolomics Profiling Reveals Beneficial Changes in Milk of Sows Supplemented with Fermented Compound Chinese Medicine Feed Additive].

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
