# Peer review of "Untargeted Metabolomics Profiling Reveals Beneficial Changes in Milk of Sows Supplemented with Fermented Compound Chinese Medicine Feed Additive"

_animals, 2022, doi:10.3390/ani12202879_

Round 1
Reviewer 1 Report
The manuscript from Zou et al. evaluated the effects of Chinese medicine feed additive to sows which altered the colostrum and mature milk metabolite components. The topic of manuscript is interesting for identifying novel feed additives to reduce postweaning diarrhea syndrome. However, the authors only reported the expected overall changes in milk metabolomics, yet without testing the effect of changes on reducing diarrhea in suckling piglets. More discussions on the findings from different metabolomics analyses are needed to improve the manuscript.
Major comments
Lack of measurements from the piglets, such as growth performance, diarrhea score, after consuming the treated milk. Such data should be easy to obtain while taking samples from the sows. The authors used analyses from chinmedomics and metabolomics, yet did not discuss the rationale and differences between the findings.
Minor comments
Line 40-41. Confusing, please rewrite.
Line 42-45. Overstated the findings in the practical application.
Line 50-78. As the major measurements are the metabolomics analysis in milks, please add the rationale and the discuss the difference of using chinmedomics and metabolomics analyses, with references.
Line 81-89. Please add the starting time point of treatment feeding to the sows.
Line 331. In discussion, please discuss the findings in different comparison designs, such as colostrum vs mature milk, etc. Why were these targeted comparisons selected? Please highlight the key findings in each comparison, especially the findings from pathway analysis in figure 2.
Line 413. The figure 3 is fine for a review paper, but not appropriate in an original research manuscript. The schematic illustration needs to be supported by the present research findings.
Line 419-421. Overstatement of the present findings.
Author Response
Major comments
Lack of measurements from the piglets, such as growth performance, diarrhea score, after consuming the treated milk. Such data should be easy to obtain while taking samples from the sows. The authors used analyses from chinmedomics and metabolomics, yet did not discuss the rationale and differences between the findings.
Response: Dear Prof., this manuscript is only to report the changes in composition of milk samples between sows supplemented with fermented compound Chinese medicine feed additive and sows supplemented without fermented compound Chinese medicine feed additive, so data such as growth performance and diarrhea are not included in this manuscript and they have been presented in the other manuscript. Chinmedomics is often used to detect the chemical compounds in Chinese herb medicine with databases of Chinese herb medicine composition, and metabolomics is used to target the compounds of non-Chinese herb medicine. The reason why we used chinmedomics is to find out what are the differences in chemical compounds of sow’s milk between experimental group and control group, because some of chemical compounds have antibacterial and anti-inflammatory functions, if the concentration of these chemical compounds can be increased in milk, it would be beneficial to the health of suckling piglets.
Minor comments
Comment No. 1: Line 40-41. Confusing, please rewrite.
Response: Dear Prof., the sentences in line 40-41 “Seven pathways (flavone and flavonol biosynthesis, galactose metabolism, phenylpropanoid biosynthesis, stilbenoid and gingerol biosynthesis, flavonoid biosynthesis, ABC transporters and purine metabolism) in colostrum and two pathways (sucrose metabolism and retrograde endocannabinoid signaling) in mature milk both from experimental group were significantly enriched respectively.” has been rewritten as follows: Seven pathways (flavone and flavonol biosynthesis, galactose metabolism, phenylpropanoid biosynthesis, stilbenoid and gingerol biosynthesis, flavonoid biosynthesis, ABC transporters and purine metabolism) in colostrum and two pathways (sucrose metabolism and retrograde endocannabinoid signaling) in mature milk were significantly enriched in experimental group compared to control group respectively.
Comment No.2: Line 42-45. Overstated the findings in the practical application.
Response: Dear Prof., “findings provide comprehensive insights into the production of functional milk for the control of diarrhea in suckling piglets by supplementing fermented compound Chinese medicine feed additive to sows” in line 42-45 has been revised to “findings of this study hint that supplementation of fermented compound Chinese medicine feed additive to sows is beneficial for the improvement of milk quality”
Comment No.3: Line 50-78. As the major measurements are the metabolomics analysis in milks, please add the rationale and the discuss the difference of using chinmedomics and metabolomics analyses, with references.
Response: Dear Prof., according to your comments, some sentences are added to the part of introduction as follows: Chinmedomics strategy is an integration of serum pharmco-chemistry of traditional Chinese medicine and “Omics” technology and often performed to determine the components of Chinese medicine [24,25], and metabolomics approach is usually applied to find out the non-Chinese medicine originated compounds [26]. In the present experiment, we applied different untargeted metabolomics approaches to characterize whether maternal feeding with fermented compound Chinese medicine feed additive has impact on the composition of active ingredients in milk.
- Wang, X.J.; Zhang, A.H.; Sun, H. Future perspectives of Chinese medical formulae: chinmedomics as an effector. OMICS. 2012, 16, 414–421.
- Zhang, A.H.; Yu, J.B.; Sun, H.; Kong, L.; Wang, X.Q.; Zhang, Q.Y.; Wang, X.J. Identifying quality-markers from Shengmai San protects against transgenic mouse model of Alzheimer's disease using chinmedomics approach. Phytomedicine.2018, 45,84-92.
- Dunn, W.B.; Broadhurst, D.; Begley, P.; Zelena, E.; Francis-McIntyre, S.; Anderson, N.; Brown, M.; Knowles, J.D.; Halsall, A.; Haselden, J.N.; Nicholls, A.W.; Wilson, I.D.; Kell, D.B.; Goodacre, R.; The Human Serum Metabolome Consortium. Procedures for large-scale metabolic profiling of serum and plasma using gas chromatography and liquid chromatography coupled to mass spectrometry. Nat. Protoc. 2011, 6, 1060-1083.
Comment No. 3: Line 81-89. Please add the starting time point of treatment feeding to the sows.
Response: the starting time point “This feeding experiment started on Aug 20, 2021” has been added before the words of “twenty pregnant crossbred sows (Landrace × Large White)…”.
Comment No. 4: Line 331. In discussion, please discuss the findings in different comparison designs, such as colostrum vs mature milk, etc. Why were these targeted comparisons selected? Please highlight the key findings in each comparison, especially the findings from pathway analysis in figure 2.
Response: Dear Prof., according to your comments, additional information are included in the part of discussion.
Comment No. 5: Line 413. The figure 3 is fine for a review paper, but not appropriate in an original research manuscript. The schematic illustration needs to be supported by the present research findings.
Response: Dear Prof., the figure 3 has been deleted according to your comments.
Comment No. 6: Line 419-421. Overstatement of the present findings.
Response: Dear Prof., the sentence of “Findings of our experiment hint that providing this functional milk to suckling piglets would be the most convenient and effective method for the control of diarrhea in suckling piglets.” has been deleted.

Reviewer 2 Report
Grammatical errors throughout - please review and edit.
Abstract - while the feed additive may have affected the milk composition, without directly evaluating piglet diarrhea, the conclusion that this supplement will help control diarrhea levels cannot be made. Should be more general, say that the compound affected milk metabolites and pathways in ways that other research suggests may be beneficial and that future research is needed to directly determine the effect of this additive on piglet diarrhea. Specify the duration of feeding the additive (from d 104 of gestation).
Introduction - much of the content in the discussion would be appropriate here. How are the metabolites and pathways evaluated related to piglet diarrhea?
Materials and methods - no discussion of experimental design (e.g., CRD/RCBD, etc.). Were data evaluated for model assumptions (normality, homogeneity of variance)? No discussion of ingredients in the feed additive. Even if the exact composition is proprietary, some general description should be provided (e.g., blend of fatty acids, essential oils, etc.).
Discussion - First sentence stating that diarrhea is the second leading cause of death needs a reference. Most sources state that the leading causes are crushing and starvation, so this statement needs to be verified. In general, the structure of the discussion could be altered for more easily following the results of this paper and their relevance. For example, stating that the specific compounds a, b, and c were increased with the feed additive and then referencing a paper that found that those compounds have antibacterial effects.
Conclusion - this conclusion states that this may be "the most convenient and effective method for the control of diarrhea in suckling piglets." Without extensive comparison to other methods for effectiveness and convenience, this statement is unfounded.
Author Response
Comment No.1: Grammatical errors throughout - please review and edit.
Response: Dear Prof., we corrected grammatical errors as we could find.
Comment No.2: Abstract - while the feed additive may have affected the milk composition, without directly evaluating piglet diarrhea, the conclusion that this supplement will help control diarrhea levels cannot be made. Should be more general, say that the compound affected milk metabolites and pathways in ways that other research suggests may be beneficial and that future research is needed to directly determine the effect of this additive on piglet diarrhea. Specify the duration of feeding the additive (from d 104 of gestation).
Response: Dear Prof., “from late pregnancy to the end of lactation” in the part of abstract has been replaced by “from d 104 of gestation to d 25 of lactation”. “findings provide comprehensive insights into the production of functional milk for the control of diarrhea in suckling piglets by supplementing fermented compound Chinese medicine feed additive to sows.” has been revised to “findings of this study hint that supplementation of fermented compound Chinese medicine feed additive to sows is beneficial for the improvement of milk quality.”.
Comment No.3: Introduction - much of the content in the discussion would be appropriate here. How are the metabolites and pathways evaluated related to piglet diarrhea?
Response: Dear Prof., we made some revisions to the parts of introduction and discussion according to your comments. To my knowledge, diarrhea of piglets during suckling period is primarily caused by the ingestion of milk with harmful microbes and inflammatory factors and by the contact of maternal feces with pathogens, the targeted metabolites in milk have antibacterial and anti-inflammatory effects, and some of metabolites can be metabolized into other antibacterial compounds via the targeted pathways, so the metabolites and pathways evaluated are thought to be related with the control of piglet diarrhea.
Comment No.4: Materials and methods - no discussion of experimental design (e.g., CRD/RCBD, etc.). Were data evaluated for model assumptions (normality, homogeneity of variance)? No discussion of ingredients in the feed additive. Even if the exact composition is proprietary, some general description should be provided (e.g., blend of fatty acids, essential oils, etc.).
Response: Dear Prof., the information of experiment design has been added to the part of “2.1. Animals and feeding” with the following words: twenty pregnant crossbred sows (Landrace × Large White) with similar body conditions and parities were randomly assigned to control group and experimental group with 10 sows (10 replicates) in each group according to a randomized complete block design. The information of ingredients in the feed additive is also added with the sentence of “The preparation, ingredients and chemical compositions of fermented compound Chinese medicine feed additive have been reported in a published paper [?].”
Comment No.5: Discussion - First sentence stating that diarrhea is the a leading cause of death needs a reference (delete “second”). Most sources state that the leading causes are crushing and starvation, so this statement needs to be verified. In general, the structure of the discussion could be altered for more easily following the results of this paper and their relevance. For example, stating that the specific compounds a, b, and c were increased with the feed additive and then referencing a paper that found that those compounds have antibacterial effects.
Response: Dear Prof., Metritis-vaginitis-mastitis of sows is prevalent in pig farm according to our surveys, and milk secreted by these sows had high levels of inflammatory factors and harmful bacteria, when piglets consume this kind of milk, diarrhea and death often occur at very high proportion. Metritis-vaginitis-mastitis of sows can decrease milk yield and diarrhea of suckling piglets can also lose appetite, so metritis-vaginitis-mastitis of sows and diarrhea of suckling piglets are the leading factors for starvation of suckling piglets. According to your comments, the sentence of “diarrhea is the second leading cause of death” in the part of discussion has been deleted. Sentences from line 346 to 356 firstly described that milk produced by sows supplemented with fermented compound Chinese medicine feed additive had higher levels of compounds with antibacterial and anti-inflammatory effects than milk produced by sows supplemented without fermented compound Chinese medicine feed additive and then followed by the referencing papers from line 357 to line 369.
Comment No.6: Conclusion - this conclusion states that this may be "the most convenient and effective method for the control of diarrhea in suckling piglets." Without extensive comparison to other methods for effectiveness and convenience, this statement is unfounded.
Response: Dear Prof., to my knowledge, control of diarrhea in suckling piglets via milk by maternal administration with Chinese herb medicine is high effective and very popular in pig farm in China, because giving medicine to control diarrhea of suckling piglets by oral administration or intramuscular injection is time and labor-consuming, so we would say “providing this functional milk to suckling piglets would be the most convenient and effective method for the control of diarrhea in suckling piglets”. Here, thank you very much for your comments, and the sentence of “Findings of our experiment hint that providing this functional milk to suckling piglets would be the most convenient and effective method for the control of diarrhea in suckling piglets.” has been deleted.

Round 2
Reviewer 1 Report
The authors have addressed all the comments. Current version of manuscript is improved and is suitable for publication.
Reviewer 2 Report
I have reviewed the revised manuscript and responses, and believe that the manuscript has been sufficiently improved to warrant publication in Animals. Thank you,